# Alveolar Epithelial Type II Cells as Drivers of Lung Fibrosis in Idiopathic Pulmonary Fibrosis

**DOI:** 10.3390/ijms21072269

**Published:** 2020-03-25

**Authors:** Tanyalak Parimon, Changfu Yao, Barry R Stripp, Paul W Noble, Peter Chen

**Affiliations:** 1Division of Pulmonary and Critical Care Medicine, Department of Medicine, Women’s Guild Lung Institute, Cedars-Sinai Medical Center, Los Angeles, CA 90048, USA; 2Department of Biomedical Sciences, Cedars-Sinai Medical Center, Los Angeles, CA 90048, USA

**Keywords:** alveolar epithelial cells, pulmonary fibrosis, epithelial cell dysfunction, stem cell exhaustion

## Abstract

Alveolar epithelial type II cells (AT2) are a heterogeneous population that have critical secretory and regenerative roles in the alveolus to maintain lung homeostasis. However, impairment to their normal functional capacity and development of a pro-fibrotic phenotype has been demonstrated to contribute to the development of idiopathic pulmonary fibrosis (IPF). A number of factors contribute to AT2 death and dysfunction. As a mucosal surface, AT2 cells are exposed to environmental stresses that can have lasting effects that contribute to fibrogenesis. Genetical risks have also been identified that can cause AT2 impairment and the development of lung fibrosis. Furthermore, aging is a final factor that adds to the pathogenic changes in AT2 cells. Here, we will discuss the homeostatic role of AT2 cells and the studies that have recently defined the heterogeneity of this population of cells. Furthermore, we will review the mechanisms of AT2 death and dysfunction in the context of lung fibrosis.

## 1. Introduction

Idiopathic Pulmonary Fibrosis (IPF) is a chronic, progressive scarring of the lungs that causes significant healthcare burden due to high morbidity and mortality [1,2,3]. It is the most common form of idiopathic interstitial pneumonia and exhibits the most severe manifestations carrying the poorest clinical outcomes [4,5,6]. Recently, two anti-fibrotic medications, Pirfenidone and Nintedanib, have been demonstrated to slow the progression of the disease and are now FDA-approved treatments for IPF [7,8,9,10,11,12,13,14,15]. Despite the progress made after decades of investigation, there is considerable room for improvement in the treatment of this incurable disease.

Pathological fibrogenesis that occurs in IPF is a dynamic process involving complex interactions between epithelial cells, fibroblasts, immune cells (macrophages, T-cells), and endothelial cells [16,17]. Early theories that chronic inflammation and repetitive damage to the alveolar epithelium promotes fibrogenesis and scar formation in the pathogenesis of lung fibrosis [18] have been largely rejected due to the lack of ongoing inflammation in IPF and the general ineffectiveness of immunosuppressive medications in curbing disease progression. In contrast, a preponderance of more recent evidence suggests that the alveolar epithelium plays a central role. Indeed, honeycombed regions of the lungs have discontinuous epithelium adjacent to hyperplastic alveolar epithelial type II cells (AT2) [19]. Accordingly, the more contemporary paradigm is that chronic injury to distal lung tissue leads to either loss or altered function of epithelial stem cells (i.e., AT2 cells), that promote dysregulated repair and pathogenic activation of fibroblasts.

Both intrinsic (e.g., genetic, aging) and environmental factors have been linked to damage of AT2 cells and contribute to the development of lung fibrosis. Tobacco smoking and viral infections have been associated with IPF [4,20]. IPF is also an age-related disease with a median age of diagnosis of 66 years old [21]. As such, several factors that accumulate with age have been found to contribute to AT2 dysfunction in IPF [22]. However, the most direct evidence supporting a role for AT2 cell dysfunction in the initiation and progression of IPF has come from the characterization of gene defects observed among patients with familial forms of the disease. Two general categories of gene mutations are observed; one category includes genes involved in the regulation of stem cell longevity, the other includes genes whose products contribute to specialized secretory functions of AT2 cells [23].

Altogether, the evidence supports a model in which AT2 injury/dysfunction serves as an early initiating event in IPF that leads to fibroproliferation and progressive loss of lung function [24]. AT2 loss can limit the ability for the repair of the damaged alveolus. Additionally, AT2 cells have maladaptive effects in the IPF lung that can drive fibrosis (Figure 1). Herein, we will review the homeostatic role of the AT2 cell and evidence for both AT2 depletion and dysfunction as contributors to IPF.

## 2. Alveolar Epithelial Type II Cells (AT2) of the Normal Mammalian Lung

The epithelium lining airspaces of the mammalian lung is maintained by regional stem and progenitor cells that are responsible for replacement of functionally specialized cell types in each compartment during homeostasis and repair [25,26]. In the alveolus, AT2 cells serve as the predominant epithelial progenitor. Lineage tracing experiments in mice show that AT2 cells defined by their expression of surfactant protein C (*Sftpc*/SFTPC) are capable of long-term self-renewal and multipotent differentiation to yield alveolar type I (AT1) cells; two characteristics that suggest either the AT2 population as a whole or a subset of AT2 cells, represent adult tissue stem cells [27]. The observation in this study that a subset of *Sftpc*-lineage positive AT2 cells exhibit greater in vivo clonogenic potential than bulk AT2 cells provides evidence that “stemness” may be a property of a subset of AT2 cells, and by inference, that AT2 cells are functionally heterogeneous. What is not clear from this work is whether AT2 cell heterogeneity is the result of differences in intrinsic potential versus microenvironmental regulation. Recent work defining a Wnt-responsive subset of AT2 cells raises the potential that a Wnt signaling microenvironment may modulate “stemness” of AT2 cells [28,29]; an appealing concept but currently not confirmed by lineage-tracing experiments. However, these data do support the notion that AT2 cells represent a heterogeneous population that includes an abundant pool of facultative progenitors and a more stem-like subset that contributes to homeostatic replacement of specialized alveolar epithelial cells.

Other epithelial stem or progenitor cells have been proposed to contribute to alveolar epithelial renewal and replacement of AT2 cells. So-called bronchio-alveolar stem cells (BASCs) were initially proposed as a multipotent “stem” cell in mouse terminal bronchioles based upon their co-expression of *Sftpc* and *Scgb1a1*, contribution to repair after chemical injury and expansion following induction of activating mutations of the *K-ras* oncogene [30]. Evidence supporting the multipotency and “stemness” of BASCs includes recent studies involving use of dual lineage-tracing strategies demonstrating that *Sftpc/Scgb1a1* dual lineage-labeled cells have potential to contribute to repair following injury to either airways or alveoli [31,32]. However, multipotency of BASCs has currently only been convincingly demonstrated in clonal culture experiments [33], with in vivo confirmation of multipotency confounded by heterogeneity within the lineage-labeled population [32]. Other candidate progenitors for replacement of AT2 cells and the alveolar epithelium include a rare subpopulation of AT1 cells [34] and basal cells that expand to repopulate damaged alveoli following influenza virus infection [35].

## 3. AT2 Depletion in Idiopathic Pulmonary Fibrosis (IPF)

A prevailing concept is that AT2 depletion potentially through repetitive microinjuries is the underlying cause of lung fibrosis [5]. Indeed, targeted deletion of AT2 cells is sufficient to induce a fibrotic response in the lungs, but it is not sustained [36,37]. Additional data supporting this idea of stem cell exhaustion is that the number of AT2 cells are diminished in IPF lungs [38,39]. However, the depletion of this vital progenitor cell is not only a function of the distorted lung architecture but may also be a precursor to fibrosis [39].

### Apoptotic Death of AT2 Cells

Mechanisms of AT2 depletion particularly in the context of early events leading to IPF are not fully elucidated. Clearly, the lung epithelium is under constant stresses as a number of studies have identified increased levels of cells undergoing apoptosis in the lungs from IPF patients, which is not seen in non-fibrotic lungs [40,41,42,43]. In fact, TGFβ, a potent pro-fibrotic cytokine, has been demonstrated to mediate its fibroproliferative effects by induction of AT2 apoptosis [44,45]. Apoptosis is a form of programmed cell death that is required as a physiological process for tissue homeostasis but can also be activated in pathological situations [46]. This regulated cell death pathway is tightly controlled at multiple levels, but if the stimulus exceeds a critical threshold, a prescribed pattern of events mediated by caspase cleavage of cellular contents to induce cell death.

In IPF lungs, the Fas-FasL pathway has several components that are upregulated in the alveolar epithelium indicating a preexisting elevation in the signals that induce apoptosis [42,47]. Similar findings were demonstrated in murine models of fibrosis [48]. Global inhibition of apoptosis with captopril or z-VAD also blunts lung fibrosis [49,50]. Additionally, blockade of Fas signaling through various pharmacologic or genetic methods attenuates whereas Fas stimulation augments lung fibrosis in bleomycin-injured mice [51,52]. Many cells that succumb to Fas-mediated death must undergo an amplification loop (type II pathway) that intensifies the death signal by crossing over to the mitochondria through the cleave of BID, a B-cell lymphoma protein-2 (BCL2) family member [53]. Further supporting the importance of the Fas-FasL pathway and apoptosis in general as an important early initiator of lung fibrosis, mice genetically deficient in *Bid* are protected from bleomycin-induced lung fibrosis [45].

## 4. ER Stress Contributes to AT2 Apoptosis

In addition to receptor-mediated activation of apoptosis, an intrinsic apoptotic pathway can also induce programmed cell death [46]. Various intracellular stresses and damage signals activate pro-apoptotic BCL-2 proteins that permeabilize the mitochondria to release mediators (e.g., cytochrome C) that create a nidus for apoptosome formation and activation of executioner caspases that cause apoptotic death [54].

Endoplasmic reticulum (ER) stress is an important initiator of AT2 apoptosis via the intrinsic pathway that has been linked to IPF [55]. The ER is an important cellular organelle that facilitates the folding and trafficking of proteins to ensure the quality control of proteins required for cellular homeostasis. In situations that overwhelm the protein folding capacity of the ER, the unfolded protein response (UPR) is activated with the aim to restore the physiological activity of the ER. Three transmembrane sensors, specifically inositol-requiring enzyme 1α (IRE1α), pancreatic endoplasmic reticulum kinase (PERK), and activating transcription factor 6 (ATF6), control the UPR [56]. When the UPR is prolonged or severe in nature, proteostasis is lost, and cells become dysfunctional and undergo apoptotic death [55].

Several studies have identified UPR activation in AT2 cells during lung fibrosis [57,58,59]. Immunohistochemical evaluation of lungs from sporadic and familial IPF demonstrated AT2 co-localization of various ER stress markers and activated caspase-3 [57,58]. Interestingly, AT2 activation of the UPR can be found in histologically normal appearing regions of IPF lungs suggesting ER stress precedes the development of fibrosis. Mice injured with bleomycin also demonstrate ER stress in AT2 cells [59]. More importantly, tunicamycin, an activator of ER stress, augments lung fibrosis [59,60], and specific activation of ER stress in AT2 cells can lead to spontaneous lung fibrosis in transgenic mice [61,62].

Several genetic variants that confer an inherited susceptibility to lung fibrosis can induce ER stress. In particular, pathologic variants of surfactant proteins, which are produced by AT2 cells, have been found to induce ER stress and augment lung fibrosis [23]. In experimental models with transgenic mice that conditionally overexpressed the *SFTPC* mutation (SFTPC^L188Q^) in AT2 cells, ER stress increased after the induction of the mutant SFTPC^L188Q^ expression [59]. Although these transgenic mice did not develop spontaneous pulmonary fibrosis, they were more susceptible to bleomycin exposure and the resultant injury-induced fibrosis. In contrast, a different *SFTPC* mutation (SFTPC^C121G^) demonstrated exaggerated ER stress, AT2 apoptosis, and development of spontaneous lung fibrosis after induction of expression in AT2 cells [61].

With aging, the lungs also become more susceptible to ER stress [55]. GRP78 is a chaperone protein that is a central regulator of ER stress by repressing IRE1α, PERK, and ATF6, the three arms that initiate the UPR. A recent study by Borok et al. demonstrated that AT2 expression of GRP78 decreases in both aged mice and in IPF lungs [63]. AT2-specific deletion of *Grp78* induced ER stress, apoptosis and lung fibrosis with an age-dependence on the severity of the effect.

A number of environmental factors that are associated with the development of IPF has also been found to induce ER stress. Smoking, which induces ER stress within the lung epithelium, is associated with an increased risk factor for acquiring IPF [64,65]. Several herpesvirus proteins were identified in AT2 cells that concomitantly showed evidence of UPR activation [20,58]. A recent study also found evidence for increased herpesvirus infections in at-risk subjects for developing interstitial lung disease [66]. Fibrosis can be augmented in mice infected with γherpesvirus [67,68,69,70]. Moreover, the viral susceptibility appears be age-related with increased AT2 ER stress and apoptosis, as well as augmented lung fibrosis in old compared to young mice infected with murine γherpesvirus [60,71].

## 5. Mitochondrial Dysfunction Causes AT2 Death

AT2 cells have the highest number of mitochondria in the lungs due to their high metabolic demands, especially during lung injury and repair [72]. Disturbance and interference of AT2 mitochondrial biogenesis, functions, and homeostasis is a known profibrotic signal [73,74,75]. The imbalance of mitochondrial dynamic due to impaired mitophagy and accumulation of mtDNA damage or irregularities of protein homeostasis leads to ER stress and program cell death of AT2 cells [74,76]. An accumulation of higher dysmorphic mitochondria (i.e., damaged mitochondria) and ER stress proteins in AT2 cells in IPF compared to control lungs suggests the significance of mitochondria dysfunction in the pathogenesis of lung fibrosis [60,77].

Mitochondrial homeostasis is regulated by integrated pathways primarily for biogenesis and recycling/disposal through mitophagy [78]. Mitofusin 1 and 2 are GTPases required for mitochondrial fusion, and AT2 deletion of either gene augmented lung fibrosis after bleomycin injury [79]. Moreover, combined AT2 deletion of both mitofusin 1 and 2 lead to spontaneous lung fibrosis. Transgenic mice expressing mitochondrial-targeted catalase (MCAT) are also protected from asbestos- and bleomycin-induced lung fibrosis mediated through the inhibition of AT2 cell mtDNA damage and apoptosis [80]. Corroborating with these data, the deficiency of sirtuin 3, a NAD-dependent deacetylase that prevents mtDNA damage, in AT2 cells promoted lung fibrosis by inducing cells apoptosis [81].

PTEN-induced putative kinase 1 (PINK1), a mitochondrial factor that facilitates mitophagy, is depleted during ER stress, with aging, and within fibrotic lungs [60,82,83]. AT2 cells in *Pink1* deficient mice are morphologically similar to AT2 cells in IPF lung, [60]. Furthermore, bleomycin-induced lung fibrosis is augmented in *Pink1* deficient mice and associated with less mitophagy, accumulation of dysmorphic mitochondria, and increased ER stress and AT2 apoptosis [60,77]. Activating transcription factor 3 (*Aft3*), a *Pink1* transcription repressor, has higher expression in fibrotic and aged lungs, suggesting this factor may account for the concomitant depletion of *Pink1* in similar conditions [60,82]. Accordingly, conditional AT2 deletion of *Aft3* protects mice from developing lung fibrosis [82]. Further demonstrating the importance of mitophagy in mediating IPF, a recent finding demonstrated a dysfunctional thyroid axis in IPF, and thyroid hormone administration increased *Pink1* levels, attenuated AT2 apoptosis, and reduced lung fibrosis via a *Pink1*-dependent mechanism [83].

A conflicting finding recently emerged where *Pgam5* deficient mice had decreased mitophagy that improved mitochondrial homeostasis, which had a protective effect during bleomycin-induced lung fibrosis [84]. Because these findings were in global *Pgam5* deficient mice, one possible explanation for the contradictory results is that mitophagy may potentially have profibrotic roles in non-AT2 compartments. Nevertheless, the majority of the evidence suggests mitophagy has a protective effect in AT2 cells and in preventing lung fibrosis [60,77,82,83].

## 6. AT2 Dysfunction in IPF

Persistent injury to the AT2 compartment not only causes a depletion of these facultative progenitor cells but can also cause irreversible alterations the capacity of these vital cells in carrying out their reparative functions. Indeed, hypertrophic and hyperplastic AT2 cells are found in the fibroblastic foci and have impaired renewal capacity [85]. Furthermore, dysfunctional AT2 cells in the fibrotic lung also produce pro-fibrotic factors that contribute to fibrogenesis [24]. Altogether, AT2 cells are not just simply depleted in the fibrotic lung as collateral damage to ongoing injury, but in addition, these cells have acquired a dysfunctional phenotype that places it as a central driver of fibrosis [37] as depicted in Figure 2.

AT2 plasticity has been best demonstrated by the ability of TGFβ to alter their cellular phenotype [86,87,88]. Although epithelial cells do not directly contribute to the mesenchymal population in IPF [89], what has become evident is that the AT2 cells develop a fundamentally altered state in the IPF lung and acquire a distinctly pro-fibrotic phenotype that promotes expansion of the mesenchymal compartment with myofibroblast activation and matrix deposition [90]. In one respect, the inability of AT2 cells to carry out proper repair of the injured alveolus can lead to scar formation [85]. In fact, AT2 cells play a central role in the activation of TGFβ, which may be self-perpetuating through the rising tension within a fibrotic lung [91]. Additionally, AT2 cells have diminished capacity for transdifferentiation into AT1 cells in the IPF lungs [5]. As such, both ER stress and telomere dysfunction, both of which have the potential to activate a program of cellular senescence, have been found to impair differentiation by stem cells [92,93,94,95,96]. Emerging evidence also demonstrates that induction of AT2 senescence in itself is sufficient to promote lung fibrosis [37].

## 7. Impaired AT2 Self-Renewal in IPF

Cellular senescence, one of the signature characterizations of the aging process, plays an important role in the pathogenesis of lung fibroproliferative disorders [97,98,99]. Fibrotic regions of lung tissue from IPF patients show both regional depletion of AT2 cells and the presence of AT2 cells with a senescent-like phenotype [37,38]. As such, AT2 senescence may have a limited reparative capacity that contributes to the fibroproliferation [22,100]. In support of this concept, senolytics show some potential as anti-fibrotics in animal models [101,102,103]. Moreover, ER stress may not only contribute to IPF through UPR-mediated AT2 death but can also impair the renewal capacity of AT2 cells by inducing cellular senescence [63]. This effect appears to be ubiquitous as the regenerative capacity of multiple types of stem cells is impaired with ER stress [92].

A major contributor to cellular senescence in lung fibrosis is telomere dysfunction [93,104]. Telomeres protect the ends of chromosomes from replicative loss through providing a mechanism for telomerase-dependent repeat expansion, thus maintaining the proliferative potential of stem and progenitor cells [105]. Shortened telomere length has been found in a subset of patients with IPF and correlates with poor survival [66,106,107]. Smoking, a risk factor for IPF, also causes a dose-dependent shortening of telomeres [108]. Similarly, telomere attrition also occurs with aging, another risk for the development of IPF [104].

A number of variants in genes that regulate telomere function has been found in IPF and could be contributing to AT2 senescence and impairment in their renewal capacity [23]. Mutations in *TERC* and *TERT*, telomerase reverse transcriptase family genes that regulate telomere length and functions, are linked to IPF [107,109,110,111,112]. A null *TERT* allele conferred a dominant transmission of disease in a familial form of pulmonary fibrosis (FIP) [109]. Risk variants of regulator of telomere elongation helicase 1 (*RTEL1*), a DNA helicase necessary for telomere stability, is also linked to FIP [113,114,115]. Subsequent studies have also found *TERT* and *RTEL1* mutations to be enriched in patients with IPF [116]. Moreover, a number of other genetic variants in *DKC1* [117], *PARN* [115,116], *NAF1* [118], and *TINF2* [119,120] have been found in FIP and associated with short telomeres. Various animal models have also demonstrated the impact of telomeres in lung fibrosis. Mice with AT2-specific *Tert* deficiency did not develop spontaneous lung fibrosis but upon bleomycin injury had more AT2 senescence and more lung fibrosis [94]. In contrast, conditional AT2 deletion of telomere repeat-binding factor 1 (TRF1), a telomere shelterin protein, caused mice to develop spontaneous lung fibrosis due to an impairment of telomere integrity [95,121].

Several non-cell autonomous factors also contribute to AT2 health and self-renewal capacity [122]. Indeed, the extracellular matrix (ECM) also has a role in the maintenance of AT2 cells [122], and these cell-matrix interactions recently were revealed to be dysfunctional in IPF, diminishing the renewal capacity of these stem cells [39]. Hyaluronan (HA) is a glycosaminoglycan that has increased abundance in the IPF ECM and can promote fibroblast invasiveness to mediate progressive lung fibrosis [123,124]. Interestingly, HA is expressed on the cell surface of normal AT2 cells and promotes stem cell renewal in a Toll-like receptor 4-dependent manner through the release of IL-6 [39]. AT2 cells also have a deficiency in HA signaling during lung fibrosis, leading to IL-6-dependent reductions in their renewal capacity [39]. Furthermore, targeted deletion of hyaluronan synthase 2 in murine AT2 cells leads to loss of progenitor cell functions and increased susceptibility to bleomycin-induced fibrosis [39].

## 8. Dysregulated Signaling Pathways Causing Intrinsic AT2 Dysfunction in IPF

Advances in single genomics have provided unprecedented opportunities to reveal changes in cellular states during normal development, tissue homeostasis, and disease. As applied to IPF patient samples, these studies have demonstrated reactivation of a series of key developmental pathways (e.g., Wnt, Hippo, Hedgehog, Notch) and the induction of altered cell states (senescence, apoptosis), that provide insights into mechanisms of altered AT2 function [38,125,126,127]. The Wnt pathway is a particularly prominent signal that is reactivated in fibrotic lungs [125,126,127,128,129,130]. Although Wnt signaling is necessary for maintaining AT2 self-renewal in development [28,29,131], the aged lung has a maladaptive response where Wnt induces cellular senescence [132,133]. As such, inhibition of Wnt signaling improves lung fibrosis in bleomycin-challenged mice [134,135,136]. One of the key AT2 functions is to transdifferentiate into AT1 cells. Because of the essential AT1 role in gas exchange, AT2 self-renewal is an important feature of to provide an endless reparative population that is ready to respond to any injury that damages the alveolar surface [137]. Wnt signaling is a developmental pathway that is activated in adult AT2 cells to drive AT1 transdifferentiation [138]. However, Wnt signaling is largely established as a nefarious event in IPF [128,129,130]. One possible way to reconcile these divergent observations is that the activated Wnt pathways are a sign of ongoing repair and regeneration of damaged alveoli [139]. Another more likely explanation is that homeostatic signals become maladapted in pathological situations with Wnt signaling causing damaging effects, such as induction of AT2 senescence [132,133].

Hippo signaling, which primarily mediates its signaling via Yes-associated protein (YAP) and transcriptional coactivator with PDZ-binding motif (TAZ), is an evolutionarily conserved pathway that crosses over with Wnt signaling [140,141]. TAZ is expressed by embryonic epithelium and is necessary for branching morphogenesis and alveolarization by promoting AT1 differentiation [142,143,144,145,146,147]. YAP/TAZ activity is an important transducer of extracellular cues that has important effects in fibrotic lung diseases [148]. In particular, fibroblasts use this mechanosensing pathway to promote a fibrotic phenotype in the fibrotic lung [149,150,151]. A prediction would be that the epithelium may also alter their phenotype in the IPF as a result of the increased stiffness within their microenvironment. Indeed, Hippo signaling is dysregulated in AT2 cells in the IPF lung to alter cell shape, proliferation, and migration as a possible contributor to lung fibrosis [152].

Notch is another developmental control pathway that regulates cell differentiation and fate through mechanisms involving lateral inhibition [153]. Upon ligand binding, Notch receptors (Notch 1 through 4) are cleaved by γ-secretase, which releases the Notch intracellular domain (NICD) that translocates to the nucleus and induces target gene (e.g., HES, HEY) transcription [154]. Although Notch has distinct roles in distal lung development, AT2 cells also require Notch signaling to coordinate crosstalk with myofibroblasts for proper alveologenesis [155,156]. Reactivation of Notch signaling has been found to be profibrotic in skin, kidney, and cardiac fibrosis [157], and Notch pathways are re-activated in AT2 cells within the IPF lungs [125,158]. Interestingly, alveolar differentiation is inhibited with persistent Notch signaling, which led to a failure to regenerate the damaged alveolus and development of honeycombed cysts that are akin to the pathology found in IPF [159]. Furthermore, Notch activation in AT2 cells induces, whereas Notch inhibition attenuates lung fibrosis [158].

## 9. Altered AT2-Fibroblast Signaling in IPF

AT2-fibroblast interactions are tightly coordinated in lung development, and this unit works in conjunction after injury to repair a damaged alveolus [29,155,156,160]. This epithelial-mesenchymal crosstalk is clearly disrupted in IPF such that AT2 cells acquire a profibrotic phenotype to aberrantly secrete profibrotic mediators as paracrine factors that stimulate and activate fibroblasts [24]. In fact, TGFβ is predominantly produced by the epithelium where the integrins needed for activation of the latent form is primarily expressed, and abrogation of TGFβ signaling within the lung epithelium can attenuate lung fibrosis [91,161,162,163,164]. CTGF is produced by the lung epithelium in response to TGFβ signaling and plays a vital profibrotic role by stimulating collagen deposition by fibroblasts [165,166,167]. AT2 cells have increased expression of CTGF in bleomycin-induced lung fibrosis, whereas blockade can abrogate fibrosis [167,168,169]. Hippo signaling (via YAP) also induces activation of target genes such as CTGF and could partially explain how the reactivation of this developmental pathway in IPF contributes to the disease [152]. Release of mtDNA from damaged AT2 cells may also contribute to fibroblast activation in IPF [76].

Hedgehog signaling is a key regulator of the epithelial-mesenchymal interactions during development and fibrosis [127,160,170]. Hedgehog is regulated by the patched surface receptors and upon ligand binding releases smoothened to activate the Gli-family transcription factors [171]. This pathway is active in the alveolar epithelium during alveolarization, after injury, and in tumors as a means to crosstalk with mesenchymal cells and coordinate their activity [152,170]. In the adult lung, hedgehog signaling controls the epithelial-mesenchymal unit by maintaining fibroblast quiescence at homeostasis and during resolution after injury [160]. Pathological reactivation of hedgehog signaling in the alveolar compartment occurs in IPF and in bleomycin-induced lung fibrosis, and pharmacologic and genetic blockade of the hedgehog epithelial-fibroblast crosstalk can attenuate experimental lung fibrosis [127,172,173,174,175,176,177,178]. In contrast, it was recently demonstrated that mice deficient in *Gli1* were not protected from lung fibrosis suggesting either *Gli2* or non-canonical hedgehog signaling may be regulating pathological effects in lung fibrosis [179].

The epithelium has defined properties that appear to keep the mesenchymal compartment in check at homeostasis [160,180,181,182,183]. Prostaglandin E2 (PGE2) levels are suppressed in IPF and have been identified as an important epithelial factor that suppresses fibroblast proliferation and activation [183,184,185]. Recently, bone morphogenic protein (BMP) was also identified to mediate a similar inhibitory effect on fibroblasts [186]. Thus, AT2 cells not only promote fibrosis through release of profibrotic mediators, but the loss of negative reinforcement factors such as PGE2 and BMP can contribute to fibroproliferation.

## 10. The Senescence-associated Secretory Phenotype Has Pro-fibrotic Effects

AT2 senescence plays a prominent role in lung fibrosis through the acquisition of a senescence-associated secretory phenotype (SASP) and is a feature of senescent cells that result in the release of a myriad of factors with inflammatory and fibrogenic properties [187]. AT2 cells within the lungs of IPF patients and bleomycin-injured mice gain features consistent with a SASP [37,38,93,125,126]. Indeed, a number of secreted inflammatory and profibrotic factors are released from AT2 cells within the fibrotic lung that can have autocrine and paracrine actions within the fibrotic microenvironment to promote lung fibrosis [24].

Several genetic variants have been found to be associated with lung fibrosis and could be mediating fibrosis through the development of the SASP in AT2 cells. Telomere-mediated senescence causes the SASP [93], and it is plausible that the various telomerase mutations associated with lung fibrosis mediate their pro-fibrotic effects by inducing a SASP in AT2 cells [107,109,110,111,112]. Hermansky–Pudlak syndrome (HPS) is an autosomal recessive disease that causes oculocutaneous albinism, bleeding diathesis, and lung fibrosis [188]. AT2 produced excessive monocyte chemotactic protein-1 (MCP-1, also known as C-C motif ligand 2 (CCL2)) that in turn promoted macrophage recruitments and TGFβ production leading to lung fibrosis [189]. This finding is consistent with the acquisition of a SASP. A *SFTPC* genetic variant associated with IPF (*SFPTC*^I73T^) also induces AT2 secretion of several chemokines, which are consistent with a SASP, to recruit Ly6C^hi^CD64^-^ monocytes [62]. Although not evaluated, this constellation of findings suggests a SASP may also be regulating the development of the spontaneous fibrosis found in this murine model.

Interestingly, a “bystander effect” is found with cellular senescence in which paracrine factors can induce neighboring cells to undergo cellular senescence [190,191,192]. Notch signaling can also promote the transfer of the senescent phenotype, which makes one posit if Notch activation in IPF may be perpetuating cell senescence in neighboring cells [193,194,195]. Extracellular vesicles (EVs) are released from cells and carry a number of factors including proteins, nucleic acids, and lipids as cargo that can be delivered to distal cells as a means of intercellular communication [196]. EVs not only change quantity and quality with cell senescence but can also play a role in causing senescence through a variety of cargo [197]. Recent evidence demonstrates increased EV production in IPF lungs could potentially be produced by senescent cells; Moreover, these EVs carry cargo such as interferon-induced transmembrane protein 3 to promote paracrine senescence [198]. EV quantify increases in the IPF lungs, and EVs isolated from fibrotic lungs can augment fibroproliferation in bleomycin-injured mice [126,199]. Wnt5a, which can induce senescence, is found on EVs from IPF lungs [132,199]. Furthermore, several microRNA cargos within EVs from IPF lungs have been found to regulate cellular senescence [126]. With the ability of EVs to carry a number of cargos that can functionally regulate distal cell phenotype, they are poised to facilitate cellular dysfunction in lung fibrosis and understanding how EVs mediate the pathological effects of the SASP could lead to interesting targets for therapeutic intervention.

## 11. Epigenetic Changes in the AT2 Promote Lung Fibrosis

The accumulation of environmental and age-related stresses over time can permanently reshape the cellular response through epigenetic changes [200]. Epigenetics refers to hereditable changes in gene expression that occur in the absence of DNA sequence alterations. Specifically, DNA methylation, histone modifications, chromatin high-order structure and remodeling, and noncoding RNAs regulate genetic accessibility to the transcriptional machinery and post-transcriptional regulation of protein translation [201]. Distinctly different patterns of DNA methylation are found in IPF patients, and multiple histone modifications have been associated with alterations in key pro-fibrotic pathways [200,202,203,204].

IPF tends to occur in older adults, and aging is an inherent risk in accumulating epigenetic changes [24,205,206,207]. IPF is more prevalent in smokers [4], and several lines of evidence have demonstrated tobacco smoke to induce long-lasting changes in gene expression, which are largely attributed to epigenetic modification of the lung epithelium [208,209,210,211,212]. Moreover, differential patterns in DNA methylation patterns can be determined by gender, which may also contribute to the increased incidence of IPF in males [213,214,215,216]. Several lines of evidence support the ability of epigenetic modulators to alter the course of lung fibrosis. Changes to the methylation state and Histone deacetylase (HDAC) inhibition by pharmacological treatment or with genetically-modified mice can blunt lung fibrosis [37,217,218,219,220,221,222]. Several noncoding RNAs have also been demonstrated to regulate lung fibrosis [200,223].

Noncoding RNAs operate through suppress of protein translation, and although without controversy, is generally considered epigenetic modifiers [201]. Moreover, multiple lines of evidence support a crossover of traditional mechanisms of epigenetic modification with microRNA expression. For example, HDAC3 controls expression of the miR-17-92 family to regulate TGFβ expression, which controls alveolar sacculation [224]. The miR-17-92 family has decreased expression in IPF and after bleomycin-induced fibrosis, whereas augmented expression after 5′-aza-2′-deoxycytidine treatment attenuated lung fibrosis [217]. The Dlk-Dio3 domain is an imprinted region that contains clusters of miRNAs and genes [225]. Methylation of the paternal intergenic germline-derived differentially methylated region (IG-DMR) represses miRNA expression from this allele such that only the miRNAs on the maternally inherited allele are transcribed [226]. Expression changes of miRNAs within the Dlk-Dio3 regions alters WNT signaling and has been associated with lung fibrosis [227,228]. Similar to the miR-17-92 family, HDAC3 controls the expression of the microRNAs within the Dlk-Dio3 region [224]. Moreover, microRNAs in the Dlk-Dio3 region have decreased expression in fibrotic conditions and can target TGFβ expression and signaling [180,224]. In particular, miR-323a-3p, which is located in the Dlk-Dio3 region, has been demonstrated to attenuate lung fibrosis in bleomycin-injured mice [180].

## 12. Conclusions

AT2 cells as a driver of IPF is an established concept supported by an abundance of evidence that demonstrates their loss and dysfunction to have a central role in fibroproliferation. Recent delineation of AT2 subsets has improved the understanding of the way these cells participate in lung development and repair after injury. What remains to be determined is how this cellular heterogeneity may have differential AT2 responses to the various environmental exposures, genetics, and age-associated changes that contribute to injury and culminate over time in a dysfunctional phenotype that shifts from reparative programs to pathogenic fibrogenesis.

## Figures and Tables

**Figure 1 ijms-21-02269-f001:**
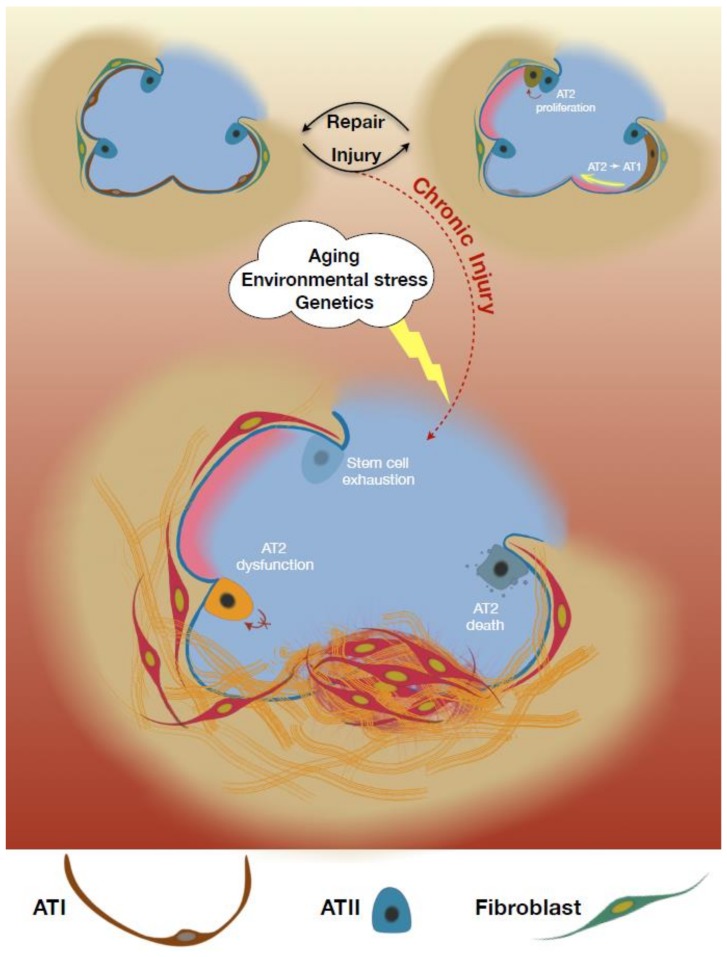
Alveolar epithelial type II cells (AT2) functions in lung fibrosis. An overview schematic illustration described functional roles of AT2 cells to maintain lung homeostasis during injuries and their profibrotic phenotypic changes that promote lung fibrosis upon the presence of risk factors e.g., aging, environmental stress, etc.

**Figure 2 ijms-21-02269-f002:**
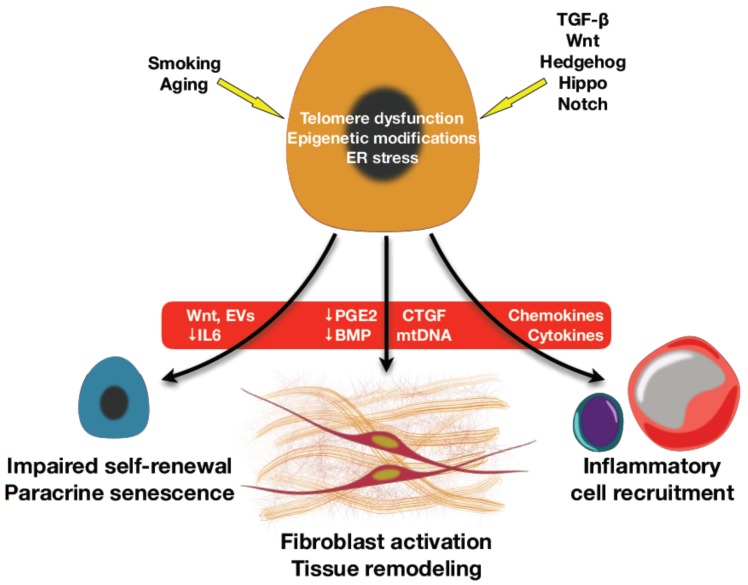
Mechanisms and regulatory signaling of alveolar type II epithelial cells (AT2) dysfunction in lung fibrosis. In lung fibrosis, environmental or intrinsic factors and several signaling regulatory pathways e.g., TGF-β, Wnt, Hedgehog, etc. stimulate AT2 senescence to release senescence-associated secretory phenotype (SASP) and other mediators that can directly activate fibroblasts and tissue remodeling. The indirect effects of SASP include inflammatory cell recruitment and AT2 self-renewal exhaustion through a paracrine effect.

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
