# Peer review of "Alveolar Epithelial Type II Cells as Drivers of Lung Fibrosis in Idiopathic Pulmonary Fibrosis"

_ijms, 2020, doi:10.3390/ijms21072269_

Round 1

Reviewer 1 Report

The manuscript "Alveolar Epithelial Type II Cells as Drivers of Lung 2 Fibrosis in Idiopathic Pulmonary Fibrosis" aimed to summarize and condense the current knowledge on AT2 "death and dysfunction" in the context of lung fibrosis. Overall, the manuscript is well written and comprehensively composed that allows easy reading for a mixed audience. 

Revising the manuscript only two minor issues were detected:

The introduction speaks of a "rising" incidence of IPF worldwide (page 1; lines 26-27). The sources the authors cited however cannot back up this statement perfectly. Hutchinson et al., Europ Resp J, 2015 does not list or show IPF with a rising incidence in their meta-analysis and Hutchinson et al., Annals ATS, 2014 discussed the global mortality but not the incidence. Please adapt the sources or wording, respectively. The chapters, " altered AT2-Fibroblast signaling in IPF" and "Dysregulated signaling pathways causing intrinsic AT2 dysfunction in IPF" discussed the impact on the "homeostatic role of AT2 cells" (also mentioned as an aim for the review on page 2, line 56). To serve the audience a complete overview on this I did not read any paragraphs on Epithelial-Mesenchymal-Transition of AT2 cells. Altghough, this is a heavily discussed topic among IPF researchers, the reader would benefit from being informed on this additional hypothesis, where the epithelium is discussed to participate directly in the ECM depletion. There are plenty papers published: i.e. Willis, duBois Borok et al., Annals ATS 2006; Goldmann et al. Resp Res, 2018 and others. Please include or comment within the text the concept of epithelial transdifferentiation the AT2 probably capable of.

Author Response

Dear reviewer # 1. Our response is attached in the PDF file.

Reviewer 2 Report

This is a very well written review discussing the role of type II epithelial cells in IPF. It is comprehensive review specific to type II cells. The manuscript would be significantly improved with a couple of figures describing some of the proposed mechanisms, particular in areas of senescence, cell death and signalling.  

Author Response

Dear Reviewer #2. Our response is enclosed as in the PDF file. 

Round 2

Reviewer 2 Report

The authors have adequately addressed the reviewer's comments.